# For Hepatocellular Carcinoma Treated with Yttrium-90 Microspheres, Dose Volumetrics on Post-Treatment Bremsstrahlung SPECT/CT Predict Clinical Outcomes

**DOI:** 10.3390/cancers15030645

**Published:** 2023-01-20

**Authors:** Crystal Seldon Taswell, Matthew Studenski, Thomas Pennix, Bryan Stover, Mike Georgiou, Shree Venkat, Patricia Jones, Joseph Zikria, Lindsay Thornton, Raphael Yechieli, Prasoon Mohan, Lorraine Portelance, Benjamin Spieler

**Affiliations:** 1Department of Radiation Oncology, Sylvester Comprehensive Cancer Center, University of Miami, 1475 NW 12th Ave, Miami, FL 33136, USA; 2Miller School of Medicine, University of Miami, 1600 NW 10th Ave, Miami, FL 33136, USA; 3Department of Radiology, Sylvester Comprehensive Cancer Center, University of Miami, 1475 NW 12th Ave, Miami, FL 33136, USA; 4Department of Medicine, Division of Digestive Health and Liver Diseases, Sylvester Comprehensive Cancer Center, University of Miami, 1475 NW 12th Ave, Miami, FL 33136, USA

**Keywords:** hepatocellular carcinoma, transarterial radioembolization, TARE, Yttrium-90, Y-90, post-TARE dosimetry, liver-directed therapy

## Abstract

**Simple Summary:**

Transarterial radioembolization (TARE) of the liver with Yttrium-90 (Y-90) microspheres is a prominent approach used to treat hepatocellular carcinoma (HCC), the most common primary liver cancer and the third-leading cause of cancer-related deaths worldwide. Recent studies have found that radiation dose estimates based on pretreatment simulations can predict HCC response to Y-90. We hypothesized that (1) Y-90 microspheres deposit heterogeneously due to variabilities in vascular dynamics; and (2) treatment response is better predicted by evaluating dose coverage of HCC in 3-dimensional space using actual Y-90 biodistribution derived from day-of-treatment nuclear imaging. We reviewed a cohort of 50 consecutive HCC patients with TARE Y-90 lobar treatments at a single institution looking for associations between volumetric dose coverage and clinical outcomes. Best treatment response most often occurred at 6 months post-TARE, with a migration toward better response after 3 months, complicating early imaging assessments. Islands of underdosed HCC appeared to compromise outcomes even when the mean or median dose to tumor was high. When prescribed dose increased along with the burden of disease, so did the mean dose to non-tumorous liver, limiting the safety of dose escalation. A multidisciplinary approach promises to accelerate advances in TARE dosimetry leading to improved clinical outcomes.

**Abstract:**

In transarterial radioembolization (TARE) of hepatocellular carcinoma (HCC) with Yttrium-90 (Y-90) microspheres, recent studies correlate dosimetry from bremsstrahlung single photon emission tomography (SPECT/CT) with treatment outcomes; however, these studies focus on measures of central tendency rather than volumetric coverage metrics commonly used in radiation oncology. We hypothesized that three-dimensional (3D) isodose coverage of gross tumor volume (GTV) is the driving factor in HCC treatment response to TARE and is best assessed using advanced dosimetry techniques applied to nuclear imaging of actual Y-90 biodistribution. We reviewed 51 lobar TARE Y-90 treatments of 43 HCC patients. Dose prescriptions were 120 Gy for TheraSpheres and 85 Gy for SIR-Spheres. All patients underwent post-TARE Y-90 bremsstrahlung SPECT/CT imaging. Commercial software was used to contour gross tumor volume (GTV) and liver on post-TARE SPECT/CT. Y-90 dose distributions were calculated using the Local Deposition Model based on post-TARE SPECT/CT activity maps. Median gross tumor volume (GTV) dose; GTV receiving less than 100 Gy, 70 Gy and 50 Gy; minimum dose covering the hottest 70%, 95%, and 98% of the GTV (D70, D95, D98); mean dose to nontumorous liver, and disease burden (GTV/liver volume) were obtained. Clinical outcomes were collected for all patients by chart and imaging review. HCC treatment response was assessed according to the modified response criteria in solid tumors (mRECIST) guidelines. Kaplan-Meier (KM) survival estimates and multivariate regression analyses (MVA) were performed using STATA. Median survival was 22.5 months for patients achieving objective response (OR) in targeted lesions (complete response (CR) or partial response (PR) per mRECIST) vs. 7.6 months for non-responders (NR, stable disease or disease progression per mRECIST). On MVA, the volume of underdosed tumor (GTV receiving less than 100 Gy) was the only significant dosimetric predictor for CR (*p* = 0.0004) and overall survival (OS, *p* = 0.003). All targets with less than CR (n = 39) had more than 20 cc of underdosed tumor. D70 (*p* = 0.038) correlated with OR, with mean D70 of 95 Gy for responders and 60 Gy for non-responders (*p* = 0.042). On MVA, mean dose to nontumorous liver trended toward significant association with grade 3+ toxicity (*p* = 0.09) and correlated with delivered activity (*p* < 0.001) and burden of disease (*p* = 0.05). Dosimetric models supplied area under the curve estimates of > 0.80 predicting CR, OR, and ≥grade 3 acute toxicity. Dosimetric parameters derived from the retrospective analysis of post-TARE Y-90 bremsstrahlung SPECT/CT after lobar treatment of HCC suggest that volumetric coverage of GTV, not a high mean or median dose, is the driving factor in treatment response and that this is best assessed through the analysis of actual Y-90 biodistribution.

## 1. Introduction

Transarterial radioembolization (TARE) with Yttrium-90 (Y-90) microspheres, also known as selective internal radiation therapy (SIRT), has become a prominent therapeutic approach for the treatment of inoperable hepatocellular carcinoma (HCC) [1]. TARE selectively targets HCC by exploiting the hepatic arterial blood supply that favors tumor over normal liver [2]. In the TARE procedure, Y-90 microspheres are delivered by guided catheterization into tumor-feeding branches of the hepatic artery. Y-90, a beta-emitting isotope with a half-life of 2.67 days and a tissue penetration range of 2.5 to 11 mm, is formulated in glass (TheraSpheres; Boston Scientific, Marlborough, MA, USA) or resin (SIR-Spheres; SIRTeX Medical, Woburn, MA, USA) beads with diameters between 20 μ and 60 μ. The short half-life, limited range, and high beta energy (average 0.93 MeV, maximum 2.26 MeV) make Y-90 a model isotope for TARE [3]. After delivery, Y-90 accumulates in the tumor microvasculature downstream from the catheter-release point, embolizing arterioles and delivering tumoricidal radiation doses [2]. For HCC patients with Child Pugh (CP)-A liver function, TARE has been shown to extend median overall survival (OS) beyond 15 months in numerous studies [4,5,6].

The TARE results could be further improved by addressing inaccuracies in the simulation technique, the prediction of Y-90 distribution, and the post-TARE assessment of Y-90 biodistribution. In the simulation phase, differences in physical properties and flow dynamics between the radioactive agent used (technetium-99m (Tc-99m)-labeled macroaggregated albumin (MAA)) and Y-90 make Tc-99m MAA mapping controversial as a reliable predictor of actual Y-90 biodistribution [7,8,9]. Several publications on the dosimetry of lobar TARE in treatment HCC, including SARAH [10,11] and TARGET [12], rely primarily on pretreatment Tc-99m MAA single photon emission computed tomography (SPECT)/computed tomography (CT) rather than post-treatment Y-90 SPECT/CT or positron emission tomography (PET)/CT for assessing actual delivered dose to tumor. Those studies correlated high tumor dose (derived from SPECT/CT of pretreatment Tc-99m MAA) with treatment response (based on analysis of follow up imaging per modified response criteria in solid tumors (mRECIST)). In defining dose, measures of central tendency were employed, in particular median or mean dose to the tumor (described as “tumor radiation-absorbed dose” [11] or “tumor absorbed dose” [12]), as representative of the dose to the entire target volume.

In the field of radiation oncology, measures of central tendency such as mean or median dose are at times used to predict the toxicity of select normal tissues (organs-at-risk or OAR) [13], but currently are considered inadequate in the assessment of the absorbed dose to the target volume [14]. In International Commission on Radiation Units and Measurements (ICRU) report No. 83, radiation oncologists are instructed to carefully inspect absorbed-dose distributions in three-dimensions to make sure that the planning target volume (PTV) is adequately irradiated [14]. Measures of central tendency alone do not capture the key parameters predictive of treatment response, such as the percentage or volume of tumor receiving less than the critical dose threshold needed for optimal tumor cell killing, nor can they identify the anatomic location of an underdosed area. In radiation oncology, volumetric coverage combined with dose volume histogram (DVH) analysis is used to interrogate brachytherapy and external beam radiotherapy (EBRT) planning for quality assurance [15]. These dosimetric techniques are rarely employed in the Y-90 space, possibly due to radiation oncology’s absence from the TARE care pathway at many institutions. For the purposes of the current study, the analysis of 3D isodose coverage and DVH are referred to as “advanced dosimetry”.

Today, commercial software is capable of converting Y-90 biodistribution extracted from post-TARE SPECT/CT or PET/CT imaging into 3D dose deposition maps using dose-point kernel convolution [16,17]. Advances in computational three-dimensional (3D) treatment planning systems (TPS) and high-resolution 3D image data permit detailed 3D dose estimations after TARE. These calculations, combined with accurate fusion to anatomic imaging, allow for the prompt application of advanced dosimetry to volumes of interest post-Y-90 SPECT/CT. Such volumetric analyses have the potential to identify areas of underdosed HCC predictive of less than complete treatment response and to influence post-TARE outcome assessment and clinical decision-making.

In this single-institution retrospective analysis of HCC patients treated with lobar Y-90 TARE, we hypothesized that volumetric coverage of the target volume is the driving factor in treatment response and is best assessed using advanced dosimetry applied to nuclear imaging of actual Y-90 biodistribution.

## 2. Methods

### 2.1. Patient Enrollment and Baseline Characteristics 

From January 2015 to January 2019, 51 lobar TARE procedures were performed with Y-90-loaded glass (TheraSpheres; Boston Scientific, Marlborough, MA, USA) or resin (SIR-Spheres; SIRTeX Medical, Woburn, MA, USA) microspheres on 43 HCC patients. The ethics committee of the University of Miami (20130430) approved the use of TARE, and written informed consent was obtained for each patient. The suitability for TARE was determined by multidisciplinary consultation involving physicians from the Liver Transplant, Hepatology, Medical Oncology, Interventional Radiology, and Radiation Oncology departments. All patients were retrospectively enrolled in an institutional review board (IRB) approved database of patients with HCC referred for TARE. A patient was considered for TARE if the patient (1) was not a candidate for imminent orthotopic liver transplantation or partial hepatectomy; (2) preserved CP-A or B liver function; (3) had no clinical or radiological evidence of extrahepatic disease; (4) maintained an Eastern Cooperative Oncology Group (ECOG) performance status score between 0 and 2; and (5) had no contraindication to angiography and visceral catheterization. 

TARE was the first-line therapy for 31 patients and the second- or third-line therapy for 12 patients with recurrent HCC previously treated with transarterial chemoembolization (TACE), microwave ablation (MWA), hepatectomy, radiofrequency ablation (RFA), or stereotactic body radiation therapy (SBRT). Thirteen (13) patients presented with locally advanced HCC, Barcelona Clinic Liver Cancer (BCLC) Stage C, and portal vein tumor thrombus (PVTT) (n = 13), but without extrahepatic spread. Twenty-six (26) patients were treated with TheraSpheres glass particles and 17 with SIRTeX resin particles. Baseline characteristics were collected for all patients by chart and imaging review and are reported in Table 1.

### 2.2. Mapping Angiography and ^99m^Tc-MAA Simulation

Patients underwent cross-sectional liver imaging with contrast enhanced triple-phase CT (ceCT) or magnetic resonance imaging (MRI) within 1 month prior to Tc-99m MAA simulation. The simulation protocol included hepatic angiography followed by Tc-99m MAA infusion, SPECT/CT imaging, and lung shunt fraction determination. The purpose of arteriography was to map hepatic vasculature, identify extrahepatic collateral vessels that might contribute to non-target embolization, and to optimize catheter position for treatment. Angiography involved the interrogation of the celiac trunk and superior mesenteric artery (SMA), as well as the sub-selective catheterization of hepatic arteries/segmental branches based on HCC location. Extrahepatic supply embolization was performed on a case-by-case basis. Before the end of the angiography procedure, Tc-99m MAA was injected in the branch of the artery perfusing the liver volume containing the targeted tumor. Every effort was made to image the patient with SPECT/CT within 1 h after Tc-99m MAA injection in order to estimate lung shunt fraction (LSF) and identify GI tract reflux. Absolute contraindications to TARE identified from simulation included pulmonary shunt fraction >20%, or GI tract reflux despite embolization. 

SPECT/CT and diagnostic imaging were transferred via a Digital Imaging and Communication in Medicine (DICOM) file to a 3D treatment planning system (MIM v. 6.9.3; MIM Software Inc., Cleveland, OH, USA) and fused to facilitate anatomic and target structure delineation. The segmentation of target volumes and liver was performed by a Radiation Oncologist specializing in TARE, and these volumes were the basis for activity calculations. The prescription dose was 120 Gy for TheraSpheres using the MIRD partition model and 85 Gy for SIRSpheres using the BSA method. Treatment activity was determined by mutual agreement among the supervising radiation oncologist, interventional radiologist, and medical physicist. Y-90 TARE was performed within 21 days of simulation and similarly involved mapping angiogram, microsphere infusion (with Y-90), and immediate post-treatment bremsstrahlung SPECT/CT. 

### 2.3. Post-Treatment 3D Advanced Dosimetry

Post-treatment Y-90 bremsstrahlung SPECT-CT data were acquired, reconstructed, registered, fused, and transferred via DICOM into MIM (See Figure 1). Bremsstrahlung imaging in SPECT/CT mode was performed using a Symbia Intevo (Siemens Healthineers; Erlangen, Germany) SPECT/CT system. Bremsstrahlung imaging using gamma camera instrumentation is challenging because it relies on a continuous spectrum of X-rays in a wide range of energies (50–400 keV) rather than a discrete photopeak energy acquisition, as with gamma emitters [18,19]. As such, there are no clear recommendations or a standardized protocol for the acquisition of Y-90 bremsstrahlung X-rays. In our institution, we follow the protocol implemented by Siman et. al. [18]. This method utilizes a primary energy window in the range of 90–125 keV with an additional energy window in the range of 310–410 keV for modeling background compensation. SPECT/CT of the chest/abdomen was performed with medium-energy collimators in a non-circular orbit, step and shoot mode with 64 azimuthian steps at 30 s/step in a 128 × 128 matrix with a 4.8 mm pixel size, followed by a low-dose non-diagnostic CT at 130 kVp, 50 effective mAs and 3.0 mm slice thickness. The SPECT and CT images were reconstructed using the manufacturer’s standard clinical software: for SPECT, ordered-subset expectation maximization (Flash 3D) was used with four iterations, four subsets, and a Gaussian pre-filter with 9.0 mm FWHM; CT images were reconstructed with filtered-back projection in a 512 × 512 matrix. The SPECT images were corrected for attenuation using a CT generated attenuation map.

The absorbed dose was obtained using SurePlan software (MIM SurePlan™) by the convolution of the activity matrix from SPECT bremsstrahlung images and the dose voxel kernel value as calculated by Monte Carlo simulation. VOI (gross tumor volume (GTV), normal liver (whole liver-GTV) were contoured, activity distributions using posttherapy bremsstrahlung were determined, and associated dose volume histograms (DVHs) were generated (Figure 2). Primary dosimetric parameters minimum, maximum, mean, and median GTV dose; minimum dose covering the hottest 2%, 5%, 50%, 70%, 95%, and 98% of the GTV (D2, D5, D50, D70, D95, D98, respectively); tumor volume receiving less than 120 Gy, 100 Gy, 70 Gy and 50 Gy; mean dose to nontumorous liver, and disease burden (GTV/liver volume) were extracted from inspection of DVH and isodose coverage of GTV and non-tumorous liver. Secondary dosimetric parameters such as activity density within non-tumorous liver (*non-tumorous liver volume/delivered activity in mCi*); heterogeneity index [20] (*HI* = (*D*5/*D*95)), conformity index [21] (*CI* = (*Tissue volume receiving* ≥ 120 *Gy*)/*GTV*), and healthy tissue overdose factor [22] (*HTOF* = (*nontumorous liver volume receiving* ≥ 120 *Gy*)/*GTV*) were derived for analysis.

### 2.4. Clinical Follow-Up, Treatment Response, and Toxicity Assessment

Post-TARE, patients were followed every 2 weeks until any acute toxicities had resolved, then monthly for 3 months to observe for radioembolization-induced liver disease (REILD) or other procedure-related adverse events. Typically, ceCT or dynamic MRI were obtained 3-months post-TARE and then at 3-month intervals. For this study, clinical outcomes were collected for all patients by chart and imaging review. Treatment response was assessed by a board certified radiologist according to the mRECIST guidelines [23]. Per mRECIST, complete response (CR) is defined as the disappearance of any intratumoral arterial enhancement in all target lesions; partial response (PR) is defined as at least a 30% decrease in the sum of diameters of viable target lesions; stable disease (SD) is defined as any case that does not qualify for either partial response or progressive disease; and progressive disease (PD) is defined as an increase of at least 20% in the sum of the diameters of viable target lesions. In this study, patients were dichotomized as CR or non-complete responders (nCR), per mRECIST as assessed on post-TARE imaging by a board-certified radiologist; or as objective responders (OR), defined as CR or PR, or non-responders (NR), defined as either SD or PD according to the same criteria. For toxicity analysis, acute toxicity was defined as toxicity arising within 90 days post-TARE. For patients who had two TARE treatments separated by >24 h, the time period for initial TARE acute toxicity evaluation ended at the time of second TARE or at 90 days, whichever came first. For patients who had split-dose TARE treatments to different targets on the same day, these were considered as a single treatment for acute toxicity analysis but were considered independently when assessing objective response. Toxicities were classified according to Common Terminology Criteria for Adverse Events (CTCAE) version 5.0, National Cancer Institute, Cancer Therapy Evaluation Program.

### 2.5. Statistics

A univariate analysis was used to identify parameters associated with treatment response, progression-free survival (PFS), overall survival (OS), toxicity ≥ grade 2, and toxicity ≥ grade 3. Parameters with *p* value ≤ 0.20 after univariate analysis were subjected to multivariate analysis (MVA). Logistic regression was used for dichotomous outcomes and Cox proportional hazards for survival time analyses. All statistical tests were two-sided except the analysis of variance for dosimetric parameters, and *p* values < 0.05 were considered statistically significant. Tumor response was grouped by the mRECIST objective response definition for responders (CR plus PR) versus non-responders (SD plus PD). The best response was used for each tumor during the study period. PFS was defined as the time between TARE and progression on imaging (CT scan or MRI) or death or second cancer. OS was defined as the time between TARE and the last follow-up visit or death. Survival rates were estimated using the Kaplan–Meier method and compared with log-rank tests. Dosimetric parameters between responders and non-responders were compared using one-way analysis of variance (ANOVA) and Fisher’s Least Significant Difference (LSD) tests. Areas under the receiver operating characteristic curve (AUC) were estimated to assess discriminatory accuracy in predicting treatment response, toxicity ≥ grade 2, and toxicity ≥ grade 3. To evaluate the consistency of findings, two commercially available software packages (MIM and DOSIsoft (DOSIsoft, Paris, France)) that perform 3D voxel-based dosimetry for Y-90 TARE were compared using correlation coefficients. All analyses were performed using STATA version 13.1 (StataCorp LLC., College Station, TX, USA) and R software (the R Foundation).

## 3. Results

Fifty unique patients and 59 consecutive TARE treatments were considered for analysis. Seven patients and their corresponding eight procedures were excluded due to the absence of available post-TARE clinical and imaging follow-up. The remaining 43 patients and their corresponding 51 treatments were included in the study. Median patient age was 67 years (range, 48–86 years), and 33 of 43 patients (77%) were male. All patients had cirrhosis, with 40 patients (93%) classified as CP-A and 3 patients (7%) as CP-B. The main risk factors for cirrhosis were viral hepatitis (58%) and alcohol use (33%). Twelve patients (28%) were BCLC-A, 18 patients (42%) were BCLC-B, and 13 (30%) were BCLC-C. Sixteen patients (37.2%) had portal hypertension noted on diagnostic imaging just prior to mapping of the angiogram. Patient baseline characteristics are shown in Table 1.

Forty-three patients underwent 51 Y-90 radioembolizations. TheraSpheres were used for 30 procedures (59%) and SIR-Spheres were used for 21 (41%). Eight patients received bilobar embolizations: two patients received bilobar TARE as a split-dose on the same-day, and 6 other patients received bilobar TARE with right and left liver lobe administrations on different days (mean interval of 54 days and range 27–95 days). Fifteen patients had solitary tumors, 26 had multifocal tumors, and 2 had diffuse infiltrative disease. Treatment volumes were the involved liver lobes prescribed to 120 Gy using TheraSpheres and 85 Gy using SIR-Spheres. The median GTV size per treatment was 85.3 cc (range 1.2–1593 cc), and the median administered activity was 39.25 mCi (range 7.52–132.16 mCi). Overall median potential clinical follow-up was 48 months (range 24–72 months). Thirty-three out of 43 patients developed locoregional or distant progression, 16 developed metastatic disease, and 33 patients had died by the time of this analysis. 

The mean time from TARE to initial mRECIST assessment was 87 days (range 40–136 days). After the initial post-TARE assessment, mRECIST scores improved on subsequent studies for 12 patients (Figure 3). In that group, five patients characterized with SD at 3-month follow-up subsequently developed CR, three patients with initial PR later were classified with CR, two patients with initial SD went on to develop PR, and two patients with PD were downstaged to SD. Mean time from TARE to best mRECIST score was 188 days (range 40–799 days). The best observed mRECIST OR was achieved after 27 of the 51 procedures (53%), with 23.5% (n = 12) demonstrating CR and 29.4% (n = 15) demonstrating PR. The 24 non-responders included 20 patients with SD (39.2%) and four with PD (7.8%). 

Median dose to GTV was a significant predictor of treatment response on univariate analysis (*p* = 0.035), with responders receiving a GTV median dose of 135 Gy (range, 17–384 Gy), compared with 85 Gy (range, 3.5–257.2 Gy) for non-responders (*p* = 0.04 by one-sided ANOVA). The probability of treatment response by median tumor dose is represented graphically in Figure 4 and shows an approximately linear relationship, with 90% tumor response probability at GTV dose > 375 Gy. However, median dose to GTV did not preserve the significance on multivariate analysis. 

On multivariate logistic regression evaluating associations with mRECIST and CR, only tumor volumes receiving less than 100 Gy preserved significance (*p* = 0.0009 and *p* = 0.0004, respectively). The median volume of tumor receiving less than 100 Gy for those achieving CR was 17 cc (range 0–81), for PR 65 cc (range 21–461), for SD 86 cc (range 22–878), and for PD 117 cc (33–770). Tumor volume itself did not correlate significantly with mRECIST (*p* = 0.09) or CR (*p* = 0.12) on UVA or MVA. GTV < 100 Gy achieved an AUC estimate of 0.80 as a negative predictor of CR on ROC analysis (see Figure 5). On multivariate logistic regression evaluating associations with objective response (CR + PR), only dosimetric parameters D70 (*p* = 0.038), D95 (*p* = 0.049), conformity index (*p* = 0.021), and HTOF (*p* = 0.020) preserved significance. D98 (*p* = 0.064) trended toward significance. Responders received a significantly higher D70 radiation dose (median = 95 Gy, range 44–295 Gy) than non-responders (median = 60 Gy, range 2–187 Gy), with *p* = 0.042 by one-sided ANOVA. 

Of note, dose heterogeneity within each tumor was evaluated using the heterogeneity index (HI), defined as D5/D95 [24]. For the 51 tumors examined, the mean HI was 7 with a standard deviation of 6.7 and a range of 0.01 to 35. To put this in perspective, HI within a target volume as a quality measure for stereotactic radiosurgery is maintained at <2 [25,26], and for other forms of external beam radiation approaches 1 [24,27]. The comparatively large heterogeneity associated with Y90 TARE reflects the challenge of achieving a critical threshold dose across an entire tumor volume using the transarterial technique and provides the underlying rationale for saturating the treatment volume with an ultra-high dose to maximize treatment effect. In this study, HI in and of itself was not a statistically significant predictor of objective or complete response on univariate or multivariate analysis.

A toxicity of at least grade 2 was observed within 3 months after the majority (53%) of TARE procedures. The most common grade 2 toxicities included hypoalbuminemia (20%), fatigue (18%), hyperbilirubinemia (16%), and ascites (14%). Ten patients (20%) developed grade 3 toxicities within 3 months, the most common being ascites (12%), hyperbilirubinemia (4%), and elevated creatinine (4%). Grade 4 toxicities (four cases of hyperbilirubinemia and one of sepsis) occurred after four of 49 procedures (8%). Other major adverse events such as biliary fistula, biloma, or postprocedural death were not observed. 

On MVA, baseline bilirubin (*p* = 0.018) and mean dose to non-tumorous (*p* = 0.09, trend) were predictive of grade 3+ toxicity, and mean dose to non-tumorous liver was highly associated with prescribed activity (*p* < 0.0001) and disease burden (*p* = 0.05). On a one-way ANOVA, there was a significant difference in baseline bilirubin levels between patients with ≥grade 3 toxicity (median 1.35, mean 2.33, range 0.7–11.1) compared to patients with <grade 3 toxicity (median 0.8, mean 0.85, range 0.2–2.3), *p* = 0.005. A combined multivariate model using only baseline bilirubin level and mean dose to non-tumorous liver achieved an AUC estimate of 0.88 as a predictor of ≥grade 3 toxicity on ROC analysis (see Figure 6). Only baseline bilirubin (*p* = 0.019) and prior local HCC treatment (*p* = 0.049) were associated with grade 2+ toxicity.

The median PFS of the entire cohort was 8.2 months (range 2–61 months). For responders, median PFS was 13.1 months (range 3.3–61 months), and for non-responders median PFS was 4.7 months, (range 2–21.6 months), with PFS curves differing significantly between responders and non-responders by log-rank (*p* = 0.006). The median OS of the entire cohort was 15.1 months. One-year OS was 63% and 2-year OS was 36% by KM estimation. On multivariate Cox regression, baseline portal hypertension identified on diagnostic radiology report (*p* = 0.017), increasing volume of tumor receiving <100 Gy (*p* < 0.003), worse mRECIST score (*p* = 0.017), and toxicity ≥ grade 3 (*p* < 0.001) were significantly associated with decreased OS. Of these, portal hypertension was the only baseline variable that achieved significance, and tumor volume receiving <100 Gy the only treatment variable. Post-TARE systemic therapy with sorafenib was significantly associated with OS on UVA but not multivariate Cox regression. For TARE responders, median OS was 22.5 months, and for non-responders the median OS was 7.6 months, with OS curves differing significantly between responders and non-responders by log-rank (*p* = 0.009). 

## 4. Discussion

Recent studies of TARE Y-90 of HCC associate increased tumor dose with improved treatment response based on the dosimetry of pretreatment Tc-99m SPECT/CT and on post-treatment Y-90 SPECT or PET/CT, and highlight the absence of dosimetric predictors of grade 3+ treatment-related toxicity [11,12,28,29,30]. In sum, the current literature points toward TARE Y-90 dose escalation as the mainstay of treatment response. The high tolerance of nontumorous liver for Y-90 dose escalation has been described elsewhere [31,32,33,34,35]. Our findings differ from existing data in important respects. In our study, median or mean absorbed dose to tumor [11,12] does not survive multivariate analysis for significance, and we believe its widespread adoption as the key predictive dosimetric parameter represents an oversimplification that obscures the clinical importance of volumetric dose coverage of tumor. In our view, the benefit of high dose prescription with Y-90 is not one based on radiobiology of HCC tumor cell killing, but on preferential vascular flow and the need to saturate the treatment volume in order to overcome the heterogeneous dose deposition inherent to the dynamics of Y-90 biodistribution. Even with macrodosing to achieve treatment volume saturation, heterogeneous and unpredictable Y-90 biodistribution risks leaving pockets of tumor beneath the volumetric threshold that correlates with poor clinical outcomes. Moreover, our data suggest limits to dose escalation in lobar treatments due to the elevated risk of grade 3+ toxicity. As prescribed activity increases along with the burden of disease, so do mean dose to non-tumorous liver and G3+ toxicity. Therefore, dose escalation is not an open-ended solution in the lobar setting. The volumetric analysis of isodose coverage is needed to assess the probability of HCC response to Y-90, a process made possible using advanced dosimetry principles borrowed from radiation oncology. 

In this study, we identified the critical dose threshold as 100 Gy, a number highly associated with mRECIST response in general, CR alone, and OS. Beyond our own findings, 100 Gy is the critical “radiation-absorbed dose” identified in the SARAH study as the determinant of enduring treatment response [11], and the biologically effective threshold dose (BED for α/β = 10) associated with enhanced local control of various liver tumors, HCC included, using stereotactic body radiotherapy (SBRT) [36,37,38,39,40,41]. It is therefore our view that Y-90 dose escalation is useful as a means to diminish the risk of leaving underdosed pockets of tumor, but does not eliminate the need for a slice-by-slice inspection of Y-90 isodose topography, as per ICRU report no. 83, intended to identify tumor cold spots that may place patients at high risk for recurrence [14]. Early identification of underdosed regions could provide interventionists with an opportunity for precise and timely retreatment with the intent to convert potential non-responders to responders, and potential partial responders to complete responders. In lobar TARE Y-90, the clinical impact of early identification and timely retreatment of probable non-responders is apparent from our data given that the OS of objective responders is >22 months vs. 7 months for non-responders. 

A tumor volume threshold of >20 cc receiving <100 Gy appears to represent underdosing significantly enough to prompt careful surveillance for possible early retreatment. According to our findings, the median volume of tumor receiving less than 100 Gy for those achieving CR was 17 cc (range 0–81), for PR 65 cc (range 21–461), for SD 86 cc (range 22–878), and for PD 117 cc (33–770). Large tumor volume itself was not predictive of less than CR (*p* = 0.12). It is unclear from this lobar dataset if scattered cold spots within the tumor have the same predictive significance for less than CR as a single or a few large areas of underdosing. We are in the process of interrogating a separate cohort of high-dose segmental treatments in which CR is a more common phenomenon in order to address this question [42]. 

Along with the obvious clinical benefits of truly predictive pretreatment dosimetry, at our institution we have committed to post-TARE analysis because Tc-99m scout dosing, due to inherent variability in physical properties and flow dynamics, cannot identify underdosed regions of HCC as accurately as dosimetry based on the actual Y-90 treatment itself. We believe that post-TARE Y-90 dosimetry has the potential to inform arguments in favor of early retreatment of HCC and also against unnecessary retreatment. Retreatment decision-making is critically important, as each additional liver-directed intervention can degrade hepatic reserve, compromising patient eligibility for effective systemic therapies [43]. As a response to this retrospective analysis, we have adopted the following algorithm after lobar TARE Y-90 at our institution:(1)Cold spots of volume >20 cc based on immediate post-TARE Y-90 dosimetric analysis trigger early imaging follow up. If there is anatomic concordance between regions of underdosing identified by dosimetry and areas of concern for residual disease in early imaging, informed by close attention to tumor marker dynamics, then these factors trigger multidisciplinary review and the discussion of early retreatment.(2)If post-TARE dosimetry identifies excellent coverage of the tumor without cold volumes of significance, then early imaging follow-up is deferred to the norm of 3 months, and even then a finding on imaging of apparent residual disease or recurrence is met with caution, as median time to best mRECIST response in our cohort was 6 months, with a range beyond a year. Radiology assessments of post-TARE imaging are often the primary determinant for or against repeat intervention. Our findings provide evidence that conventional radiology at 3 months can miss a migration toward treatment response 30% of the time (12 out of 41 patients migrated toward OR after initial post-treatment imaging study). mRECIST guidelines remain the standard for the assessment of HCC response to local and systemic therapy [44], but volumetric dosimetry appears capable of supplementing and refining its utility in the post-TARE setting.

If retreatment is adopted, safety profiles of various interventions, from local ablation to repeat TARE to SBRT, remain undefined in the context of very early retreatment. In the external beam radiation setting, the risk of liver toxicity increases with multiple courses of radiation [45]. Repeat TARE to residual or recurrent tumor has been investigated both for resin and glass microspheres, and the data regarding safety is mixed [46,47,48]. Lam et al. [49] advised caution with repeat treatment due to the elevated incidence of REILD, but studies by Badar et al. [47] and Masthoff et al. [48] reported that repeat treatment to the same arterial territory was as safe as initial TARE. In the studies conducted by Lam et al. and Masthoff et al., the mean interval between initial and repeat treatments was at least 9 months, exceeding the timeframe for therapy suggested by our findings. 

Meanwhile, a recent retrospective study of patients who received TARE followed by SBRT reported acceptable tolerability and efficacy with no obvious increase in toxicity [49]. Thirty-one patients received SBRT after segmental TARE (18 to the same lesion) with a mean interval between interventions of 6.4 months (range 0.8–28). In the post-TARE SBRT cohort, the incidence of grade ≥3 toxicity was 9%, all occurring in the acute setting (≤60 days). There was no association between TARE-SBRT time interval and toxicity, and no association with toxicity if SBRT was targeting the same lesion as TARE or a different lesion. There was also no difference in CP score or ALBI score among patients treated with SBRT post-TARE vs. post-TACE. These findings suggest that early adjuvant SBRT to underdosed HCC after segmental TARE may be acceptably tolerated and clinically effective. However, this safety profile may not be generalizable to a cohort requiring lobar administrations. 

Assuming that prompt retreatment is feasible, then accurate localization, size estimation of cold regions(s), and transarterial access are factors critical to the selection of the appropriate intervention. The shape of the potential retreatment volume may vary based on the spatial resolution characteristic of post-Y-90 nuclear imaging platforms, PET vs. SPECT [50]. Although post-Y-90 SPECT imaging (used exclusively in this study) is more readily accessible than PET, SPECT limitations include inferior spatial resolution, wide energy distribution, and high scatter from bremsstrahlung photons leading to high noise outside of the body. The quantitative analysis between PET and SPECT has found less scatter and superior spatial resolution with Y-90 PET imaging [50]. To further this gap, recent data has shown that SiMP-based PET/CT systems lead to more accurate results for quantitive Y-90 measures [51]. In our study, cold regions were often irregularly shaped and multifocal, a pattern even more likely to be identified on higher resolution PET. For patchwork retreatments, ablative interventions such as MWA and RFA do not appear ideal, whereas SBRT seems to offer an attractive, non-invasive alternative. Accounting for variations in spatial resolution with appropriate security margins, SBRT is capable of this type of complex and irregular targeting, especially when applied from MRI-guided platforms using advanced motion management techniques [38]. 

The limitations of this study include a small heterogeneous sample size, 12 pretreated livers, patients lost to follow-up, and a retrospective single institution approach. The sample size may have been too small to detect meaningful statistical associations. Confounding factors such as post Y-90 systemic therapy may have played a role in overall survival estimates. For patients who received split dose TARE to separate target volumes, the target volumes were considered independently when assessing the objective response. Assuming the independence of multifocal HCC in response to radioembolization ignores the possibility of bystander or abscopal effects. Post-treatment dosimetry was dependent on the coregistration of baseline SPECT/CT imaging with post-TARE SPECT/CT, and despite best efforts, small misalignments may have occurred. To ensure the consistency of our findings, two commercially available software packages (MIM and Dosisoft) that perform 3D voxel-based dosimetry for Y-90 SIRT studies were compared for all cases in this report [52]. The systems were compared over a range of dose levels, including GTV maximum dose, mean dose, D95, D70, D50, and D2 derived from dose volume histograms. The correlation coefficients between the datasets exceeded 0.96 for all parameters assessed, indicating that the software platforms can be utilized clinically with a degree of confidence that they provide very similar output. 

Finally, cooperation between interventional radiology and radiation oncology in the TARE Y-90 care pathway promises to accelerate advances in TARE dosimetry, leading to improved patient outcomes through an integrated, multidisciplinary approach. 

## 5. Conclusions

Dosimetric parameters derived from the retrospective analysis of post-TARE Y-90 bremsstrahlung SPECT/CT after lobar treatment of HCC suggest that isodose coverage of tumor, not high median dose, is the driving factor in treatment response and that this is best assessed through the 3D analysis of actual Y-90 biodistribution. As prescribed activity increases along with the burden of disease, so does the mean dose to non-tumorous liver and G3+ toxicity, limiting the safety of dose escalation. A tumor volume threshold of >20 cc receiving <100 Gy, irrespective of median dose to the whole tumor, appears to represent underdosing significant enough to prompt careful surveillance for possible early retreatment. In select patients, early retreatment through spatial cooperation with other liver-directed therapies, informed by post-TARE Y-90 dosimetry, may upgrade treatment response and significantly impact OS for HCC patients.

## Figures and Tables

**Figure 1 cancers-15-00645-f001:**
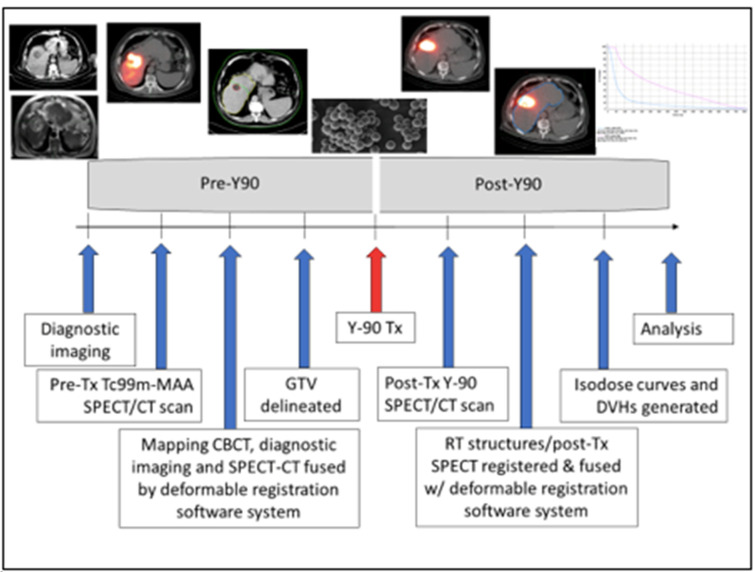
TARE Y-90 dosimetry workflow.

**Figure 2 cancers-15-00645-f002:**
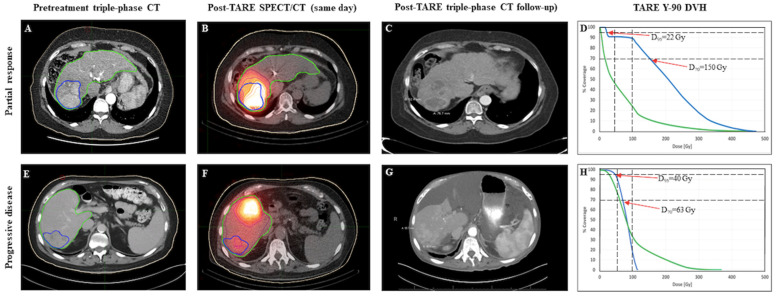
Cases of partial response (PR; **top row**) and progressive disease (PD; **bottom row**). Gross tumor volume is delineated in blue and liver volume in green on pretreatment diagnostic imaging (Panel (**A**,**E**)) and post-Y90 same day SPECT (Panel (**B**,**F**)). Follow up imaging shows areas of necrosis within the treated volume for the responder (Panel (**C**)), and new tumor growth in the non-responder (Panel (**G**)).

**Figure 3 cancers-15-00645-f003:**
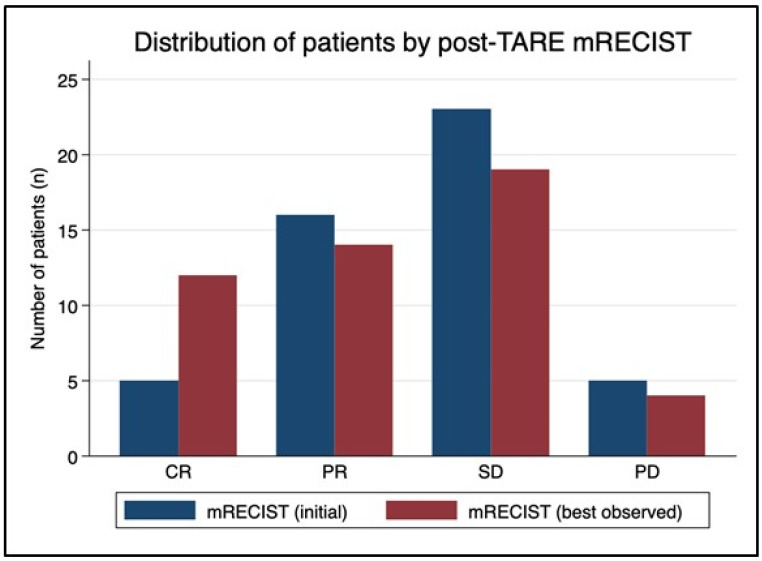
mRECIST migration over time. The mean time from TARE to initial mRECIST assessment was 87 days (range 40–136 days). After the initial post-TARE assessment, mRECIST scores improved in subsequent studies for 12 patients. In that group, five patients characterized with stable disease (SD) at 3-month follow-up subsequently developed a complete response (CR), three patients with initial partial response (PR) were later classified with CR, two patients with initial SD went on to develop PR, and two patients with progressive disease (PD) were downstaged to SD. Mean time from TARE to best mRECIST score was 188 days (range 40–799 days). Ultimately, mRECIST objective response was achieved after 27 of the 51 procedures (53%), with 23.5% of patients demonstrating CR and 29.4% demonstrating PR.

**Figure 4 cancers-15-00645-f004:**
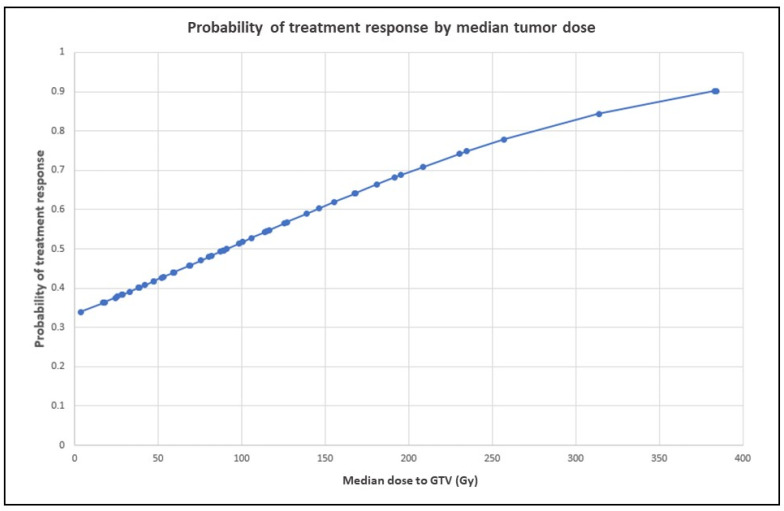
Probability of treatment response by median tumor dose. The graph shows an approximately linear relationship, with 90% tumor response probability at GTV dose > 350 Gy. Probabilities at various levels of the independent variable were derived from logistic regression odds ratios.

**Figure 5 cancers-15-00645-f005:**
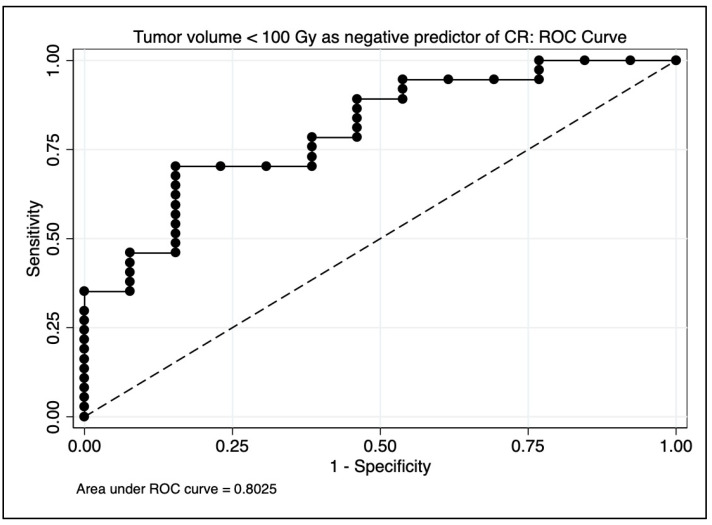
Area under the curve (AUC) estimate for predicting complete response on receiver operating characteristic (ROC) analysis. A model using only tumor volume receiving <100 Gy achieved an AUC estimate of 0.80.

**Figure 6 cancers-15-00645-f006:**
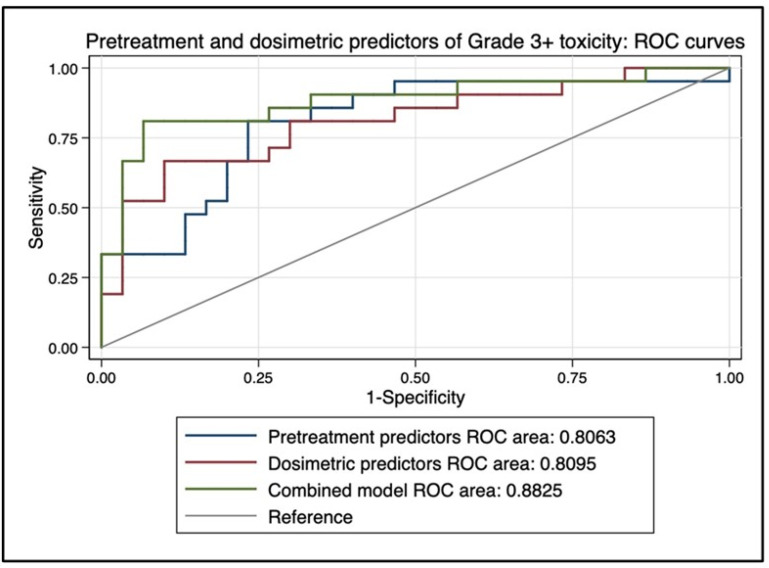
Area under the curve (AUC) estimate for predicting complete response on receiver operating characteristic (ROC) analysis. A combined multivariate model using only baseline bilirubin level and mean dose to non-tumorous liver achieved an AUC estimate of 0.88 as a predictor of ≥grade 3 toxicity.

**Table 1 cancers-15-00645-t001:** Patient baseline characteristics.

Clinical Variable	N	%	Clinical Variable		N	%
Patients	43	100	All		43	100
Gender			Child Pugh score			
Female	10	23.3	A		40	93.0
Male	33	76.7	B		3	7.0
Ethnicity			BCLC Stage			
Asian	1	2.3	A		12	27.9
Black	5	11.6	B		18	41.9
Hispanic	10	23.3	C		13	30.2
Multiracial	1	2.3	Prior RFA			
White	26	60.5	No		40	93.0
Alcohol			Yes		3	7.0
No	29	67.4	Prior SABR			
Yes	14	32.6	No		41	95.3
HCV			Yes		2	4.7
No	23	53.5	Prior TACE			
Yes	20	46.5	No		35	81.4
NASH			Yes		8	18.6
No	31	72.1	Prior resection			
Yes	12	27.9	No		41	95.3
Hemochromatosis			Yes		2	4.7
No	41	95.3	Prior MWA			
Yes	2	4.7	No		39	90.7
Tumor distribution			Yes		4	9.3
Diffuse	2	4.7	ECOG score			
Multifocal	26	60.5	0		22	51.2
Unifocal	15	34.9	1		20	46.5
Portal HTN on imaging			2		1	2.3
Unknown	1	2.3				
Absent	26	60.5	TARE treatments		51	100
Present	16	37.2	Theraspheres		30	58.8
Portal vein tumor thrombus			SIRSpheres		21	41.2
No	29	67.4				
Yes	14	32.6				
**Clinical Variable**	**Median**		**Minimum**	**Maximum**	**Std Dev**	
age	67		48	86	7.95	
AFP	58.2		1.6	60,500	13,020	
albumin	3.9		1.3	4.7	0.56	
bilirubin	0.9		0.20	2.6	0.54	
INR	1.13		0.93	2.1	0.19	
creatinine	0.90		0.39	5.6	0.83	
MELD score	6.5		0.64	16.8	4.11	

*N* number; *HBV* hepatitis B virus; *HCV* hepatitis C virus; *NASH* nonalcoholic steatohepatitis; *BCLC* Barcelona clinic liver cancer; *RFA* radiofrequency ablation; *SABR* stereotactic ablative radiotherapy; *TACE* transarterial chemoembolization; *MWA* microwave ablation; *ECOG* Eastern Cooperative Oncology Group; *TARE* transarterial radioembolization; *AFP* alpha-fetoprotein; *INR* international normalized ratio; *MELD* model for end-stage liver disease; *Std Dev* Standard Deviation; *HTN* Hypertension.

## Data Availability

Research data are stored in an institutional repository and will be shared upon request to the corresponding author.

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
