# Peer review of "For Hepatocellular Carcinoma Treated with Yttrium-90 Microspheres, Dose Volumetrics on Post-Treatment Bremsstrahlung SPECT/CT Predict Clinical Outcomes"

_cancers, 2023, doi:10.3390/cancers15030645_

Round 1
Reviewer 1 Report
In this study, the authors retrospectively analyzed HCC patients treated with lobar Y-90 TARE, to evaluate whether volumetric coverage of the target volume is the driving factor in treatment response and if it is best assessed using advanced dosimetry applied to nuclear imaging of actual Y-90 biodistribution. Y-90 dose distributions were calculated using the Local Deposition Model based on post-TARE SPECT/CT activity maps. Median gross tumor volume (GTV) dose; GTV receiving less than 100 Gy, 70 Gy and 50 Gy; minimum dose covering the hottest 70%, 95%, and 98% of the GTV (D70, D95, D98). Clinical outcomes were collected for all patients by chart and imaging review. HCC treatment response was assessed according to the modified response criteria in solid tumors (mRECIST) guidelines.
They found that median survival was 22.5 months for patients achieving objective response (OR) in targeted lesions (complete response (CR) or partial response (PR) per mRECIST) vs. 7.6 months for non-responders (NR, stable disease or disease progression per mRECIST). On MVA, the volume of underdosed tumor (GTV receiving less than 100 Gy) was the only significant dosimetric predictor for CR (p=0.0004) and overall survival (OS, p=0.003). The median volume of underdosed tumor for targets achieving CR (n=12) was 17 cc (range 0-81), and all targets with less than CR (n=39) had more than 20 cc of underdosed tumor. D70 (p=0.038) correlated with OR, with mean D70 of 95 Gy for responders and 60 Gy for non-responders (p=0.042). On MVA, mean dose to nontumorous liver trended toward significant association with grade 3+ toxicity (p=0.09) and correlated with delivered activity (p<0.001) and burden of disease (p=0.05). Dosimetric models supplied area under the curve estimates of > 0.80 predicting CR, OR, and ≥ grade 3 acute toxicity. They concluded that dosimetric parameters derived from the analysis of post-TARE Y-90 bremsstrahlung SPECT/CT after lobar treatment of HCC, suggest that volumetric coverage of GTV is the driving factor in treatment response and that this is best assessed through analysis of actual Y-90 biodistribution. As prescribed activity increases along with the burden of disease, so do mean dose to non-tumorous liver and G3+ toxicity, limiting the safety of blind dose escalation. A GTV threshold of > 20 cc receiving < 100 Gy, irrespective of median dose to the whole tumor, appears to represent underdosing significant enough to prompt careful surveillance for possible early retreatment.
Early retreatment through spatial cooperation with other liver-directed therapies, informed by post-TARE Y-90 dosimetry, has the potential to upgrade treatment response and significantly impact OS for HCC in selected patients.
The study is of clinical impact and well presented. In my opinion, the authors should further discuss the impact of liver function on the outcome of patients who underwent Y-90, as well as the potential risk of liver function deterioration after repeated transarterial treatments that might produce a shift of liver function from Child-Pugh A to Child-Pugh B thus precluding further HCC treatments such as sequential systemic therapies as previously suggested (Non-transplant therapies for patients with hepatocellular carcinoma and Child-Pugh-Turcotte class B cirrhosis. Lancet Oncol. 2017 Feb;18(2):e101-e112; Experience with regorafenib in the treatment of hepatocellular carcinoma. Therap Adv Gastroenterol. 2021 May 28;14:17562848211016959).
Author Response
We thank you for considering our manuscript entitled, “Hepatocellular Carcinoma Treated with Yttrium-90 Microspheres, Dose Volumetrics on Post-Treatment Bremsstrahlung SPECT/CT Predict Clinical,” for publication in Cancers. We thank the reviewers for taking the time to review our manuscript and for their comments and suggestions. We believe that our modifications in response to these comments have significantly enhanced the clarity and flow of the manuscript. We have addressed the reviewer comments below on a point by point basis. Our responses follow the reviewer comments while text changes are in quotes. Within the manuscript, the changes have been tracked and highlighted using red underlined text.

Reviewer 2 Report
In this retrospective study, the authors evaluated the impact of Dose Volumetrics on Post-Treatment Bremsstrahlung SPECT/CT on clinical outcome of HCC patients treated with glass or resin 90Y-microspheres. The authors found that the volume of underdosed tumor (GTV receiving less than 100 Gy) was the only significant predictor of complete response and overall survival. In addition, dose delivered to non-tumor parenchyma resulted correlated with grade 3 toxicity.
The paper is original, well-written and thoroughly documented.
Some considerations:
- abstract should be shortened to become easier to be read;
- in the introduction and in the discussion, the authors correctly mention 90Y-PET as a tool for post-hoc dosimetry. They should briefly discuss why 90Y, which is traditionally considered a pure beta-emitter, allows to acquire PET/CT scan, also citing some relevant references in the filed, such as DOI: 10.1097/MNM.0000000000001395 and DOI: 10.1186/s13550-017-0341-9;
- Although SPECT/CT is a widespread approach, its spatial resolution is absolutely inferior with respect to that of 90Y-PET/CT, and this gap is going to become even deeper with the implementation of SiMP-based detectors. Please briefly mention this in the Discussion (ref: DOI: 10.1002/mp.15880);
- for the calculation of prescribed activity, the authors employed BSA method, that is an empirical method, far from the international recommendations, and this issue should be stressed by the authors as a limitation of the study (doi: 10.1007/s00270-022-03215-x);
- In the Discussion, the authors state “The risk of liver toxicity increases with multiple courses of radiation [39] and repeated TARE is suspected of carrying increased risk for REILD, especially when treatment territories are large [40]”. I partially agree with this sentence. First of all, ref. n. 39 (Predictors of Liver Toxicity Following Stereotactic Body Radiation Therapy for Hepatocellular Carcinoma...) deals with external radiation therapy and I warmly suggest to delete it. On the other hand, the lack of consensus on REILD definition makes difficult to accurately estimate how repeated TARE procedures might have an impact on procedure-related toxicity (please cite and discuss doi:10.21037/atm-20-2658).
Some typo errors, such as “residua and recurrence”, that should be amended.
Author Response

(The authors gave the same response as above.)
